# (More than) doubling down: Effective fibrinolysis at a reduced rt-PA dose for catheter-directed thrombolysis combined with histotripsy

**Samuel A. Hendley**[1], **Aarushi Bhargava**[2], **Christy K. Holland**[3,4], **Geoffrey D. Wool**[5], **Osman Ahmed**[2], **Jonathan D. Paul**[6], **Kenneth B. Bader**[1,2]*

1 Committee on Medical Physics, University of Chicago, Chicago, Illinois, United States of America,
2 Department of Radiology, University of Chicago, Chicago, Illinois, United States of America, 3 Department of Internal Medicine, University of Cincinnati, Cincinnati, Ohio, United States of America, 4 Department of Biomedical Engineering, University of Cincinnati, Cincinnati, Ohio, United States of America, 5 Department of Pathology, University of Chicago, Chicago, Illinois, United States of America, 6 Department of Medicine, University of Chicago, Chicago, Illinois, United States of America

* baderk@uchicago.edu

**Data Availability Statement:** Data files associated with this study are accessible via Figshare (DOI: 10.6084/m9.figshare.c.5612018.v1).

## Abstract

Deep vein thrombosis is a major source of morbidity and mortality worldwide. For acute proximal deep vein thrombosis, catheter-directed thrombolytic therapy is an accepted method for vessel recanalization. Thrombolytic therapy is not without risk, including the potential for hemorrhagic bleeding that increases with lytic dose. Histotripsy is a focused ultrasound therapy that generates bubble clouds spontaneously in tissue at depth. The mechanical activity of histotripsy increases the efficacy of thrombolytic therapy at doses consistent with current pharmacomechanical treatments for venous thrombosis. The objective of this study was to determine the influence of lytic dose on histotripsy-enhanced fibrinolysis. Human whole blood clots formed *in vitro* were exposed to histotripsy and a thrombolytic agent (recombinant tissue plasminogen activator, rt-PA) in a venous flow model perfused with plasma. Lytic was administered into the clot via an infusion catheter at concentrations ranging from 0 (control) to 4.54 μg/mL (a common clinical dose for catheter-directed thrombolysis). Following treatment, perfusate samples were assayed for markers of fibrinolysis, hemolysis, and intact red blood cells and platelets. Fibrinolysis was equivalent between the common clinical dose of rt-PA (4.54 μg/mL) and rt-PA at a reduction to one-twentieth of the common clinical dose (0.23 μg/mL) when combined with histotripsy. Minimal changes were observed in hemolysis for treatment arms with or without histotripsy, potentially due to clot damage from insertion of the infusion catheter. Likewise, histotripsy did not increase the concentration of red blood cells or platelets in the perfusate following treatment compared to rt-PA alone. At the highest lytic dose, a refined histotripsy exposure scheme was implemented to cover larger areas of the clot. The updated exposure scheme improved clot mass loss and fibrinolysis relative to administration of lytic alone. Overall, the data collected in this study indicate the rt-PA dose can be

**Funding:** This work was funded by the National Heart, Lung, and Blood Institute (https://www.nhlbi.nih.gov/), Grant Number R01 HL133334. The funders had no role in study design, data collection and analysis, decision to publish, or preparation of the manuscript.

**Competing interests:** O. A. has acted as a consultant for Inari Medical, Boston Scientific, and received research grants from Inari Medical, Canon Medical, and Philips. He acted as a speaker and received compensation for Argon Medical, Canon Medical, Penumbra, Philips, and Johnson and Johnson. G.D.W. received honoraria and serves on the advisory board for Diagnostica Stago. This does not alter our adherence to PLOS ONE policies on sharing data and materials.

reduced by more than a factor of ten and still promote fibrinolysis when combined with histotripsy.

## Introduction

Deep vein thrombosis (DVT) obstructs venous blood return, with the most clinically relevant presentation in lower extremities [1]. Approximately 5% of the American population will be diagnosed with DVT in their lifetime [2], and many more cases are projected to go undiagnosed [3]. Besides the primary venous occlusion, post thrombotic syndrome, ulcerations, and phlegmasia cerulea dolens associated with DVT contribute to patient morbidity [4]. Pulmonary embolism caries a 25% mortality rate and a 30-day survival rate of 59% in high risk cases, and represents the worst possible outcome for DVT patients [5]. Proximal iliofemoral thrombi are the most likely presentation of DVT to embolize and require intervention to minimize potential vascular sequalae. Catheter-directed thrombolysis, the primary intervention to achieve recanalization for DVT [6], promotes dissolution of fibrin structure via intra-thrombus administration of the lytic agent recombinant tissue-plasminogen activator (rt-PA) [7, 8]. This approach has potential drawbacks, including hemorrhagic complications that scale with the lytic dose and treatment duration [9]. Thrombi phenotypes resistant to lysis are also prevalent in patients presenting with DVT [10], necessitating the development of a more effective treatment strategy. Adjuvant therapies that improve the outcomes for venous thrombosis patients are therefore an active area of research.

Pharmacomechanical therapies combine thrombolysis with mechanical fragmentation to achieve recanalization [11]. Sonothrombolysis relies on the ultrasound energy to improve the activity of rt-PA [12]. Ultrasound can be administered intravascularly (e.g. EKOS Endovascular System, Boston Scientific, Marlborough, MA [13] and forward facing catheters [14]) or extracorporeally [15, 16]. Bubble activity is the primary catalyst to promote clot degradation [12]. Most sonothrombolysis schemes employ exogenous agents such as microbubbles, nanobubbles, or droplets to nucleate bubble activity [17, 18]. These methods have a limited thrombus penetration profile [19] and do not sustain bubble activity for extended periods ($< 30$ s) [20, 21]. Histotripsy is a focused ultrasound therapy that achieves tissue ablation via the mechanical activity of bubble clouds [22–24]. The tension of the histotripsy pulse is sufficient to activate nanoscale nuclei that permeate soft tissues such as clot [25, 26], enabling bubble generation throughout the thrombus burden. The duty cycle for histotripsy pulsing schemes is very low ($< 1\%$), making tissue heating effects negligible [27]. Vascular structure is resistant to low levels doses of histotripsy compared to thrombus, which is ideal for DVT interventions [28]. In addition to its ablative qualities, histotripsy bubble activity induces vigorous fluid mixing to promote the delivery of therapeutics [29, 30]. Indeed, there is a growing body of literature which demonstrates synergistic enhancement of clot dissolution when histotripsy is combined with rt-PA [31–33].

To date, histotripsy-enhanced thrombolysis has been investigated for rt-PA doses consistent with current clinical pharmacomechanical approaches. A reduction in the lytic dose will potentially reduce the frequency of adverse events [9, 34]. Prior histotripsy-enhanced thrombolysis studies modeled a systemic rt-PA infusion, whereas catheter-directed therapy is the standard-of-care for DVT [6]. There are multiple differences between systemic and catheter-directed therapies that may alter treatment outcomes, including the lytic profile and administration rate. Further, the influence of the catheter on histotripsy bubble nucleation and

dynamics is unknown. In this study, the influence of rt-PA dose on catheter-directed thrombolytic efficacy combined with histotripsy was assessed *in vitro*. The change in clot mass was used as the metric of overall treatment efficacy. Assays were conducted to quantify mechanical and fibrinolytic degradation of the clot. Samples were examined histologically to assess the baseline clot structure and its interaction with the infusion catheter. Finally, acoustic emissions were tracked to quantify the strength and location of bubble activity generated within the clot during histotripsy exposure.

## Methods and materials

### In vitro clot model

Clots were formed following an established Internal Review Board-approved protocol (University of Chicago IRB #19–1300). Whole blood was drawn from 14 volunteer patients undergoing procedures at the University of Chicago Medicine cardiac or interventional radiology catheterization laboratories. Immediately after collection, 4 mL aliquots of whole blood were transferred to borosilicate test tubes of 16 mm diameter (#2378T43, McMaster-Carr, USA), and incubated at 37 ˚C in a water bath for 3 hours. At the conclusion of incubation period, clots were retracted from the container walls and significant serum was observed in the pipette. Clots were stored at 4 ˚C for 3 days to allow for full retraction [35], and were used within 2 weeks of formation. The nominal clot diameter was 10 mm. Clots formed under this protocol are uniformly homogenous with erythrocytes [32], which may not reflect the heterogeneous pathologies observed for aged *in vivo* specimen [10].

### Preparation of human plasma and recombinant tissue-type plasminogen activator

Human fresh-frozen plasma was obtained from a blood bank (Vitalant, Chicago, IL), thawed at 37 ˚C, divided into 35 mL aliquots, and stored at -80 ˚C prior to use. Recombinant tissue plasminogen activator was reconstituted according to manufacturer instructions to 1 mg/mL (Activase, Genentech, San Francisco, CA, USA), transferred to 0.5 mL aliquots, and stored at -80 ˚C. The enzymatic activity of rt-PA stored in this capacity is stable for at least seven years [36].

### Histotripsy insonation

Histotripsy pulses were generated by an eight-element, spherically-focused transducer with an elliptical geometry (9 cm major axis, 7 cm minor axis) and 1.5 MHz fundamental frequency. The transducer was driven by a custom-designed class D amplifier and matching network [37]. The transducer was designed to target femoral DVT [38], and had a focal length of 6 cm with -6-dB focal dimensions of 4.27 mm x 0.66 mm x 0.70 mm (axial direction, major axis, and minor axis, respectively). An ultrasound probe (L11-5v, Verasonics, Inc., Kirkland, WA, USA) was inserted through a central opening in the transducer for image guidance and assessment of bubble activity.

### Experimental procedure

To model venous obstruction, clots were sectioned to 1 cm length and introduced into the flow model depicted in Fig 1. The model consisted of a syringe pump (EW-74900-20, Cole-Parmer, Vernon Hills, IL, USA) that drew thawed plasma through a model vessel of latex tubing 12.7-mm-inner-diameter and 1.6-mm wall thickness (McMaster-Carr, Elmhurst, IL, USA). The plasma flow rate was 0.65 mL/min, consistent with slow flow observed in DVT

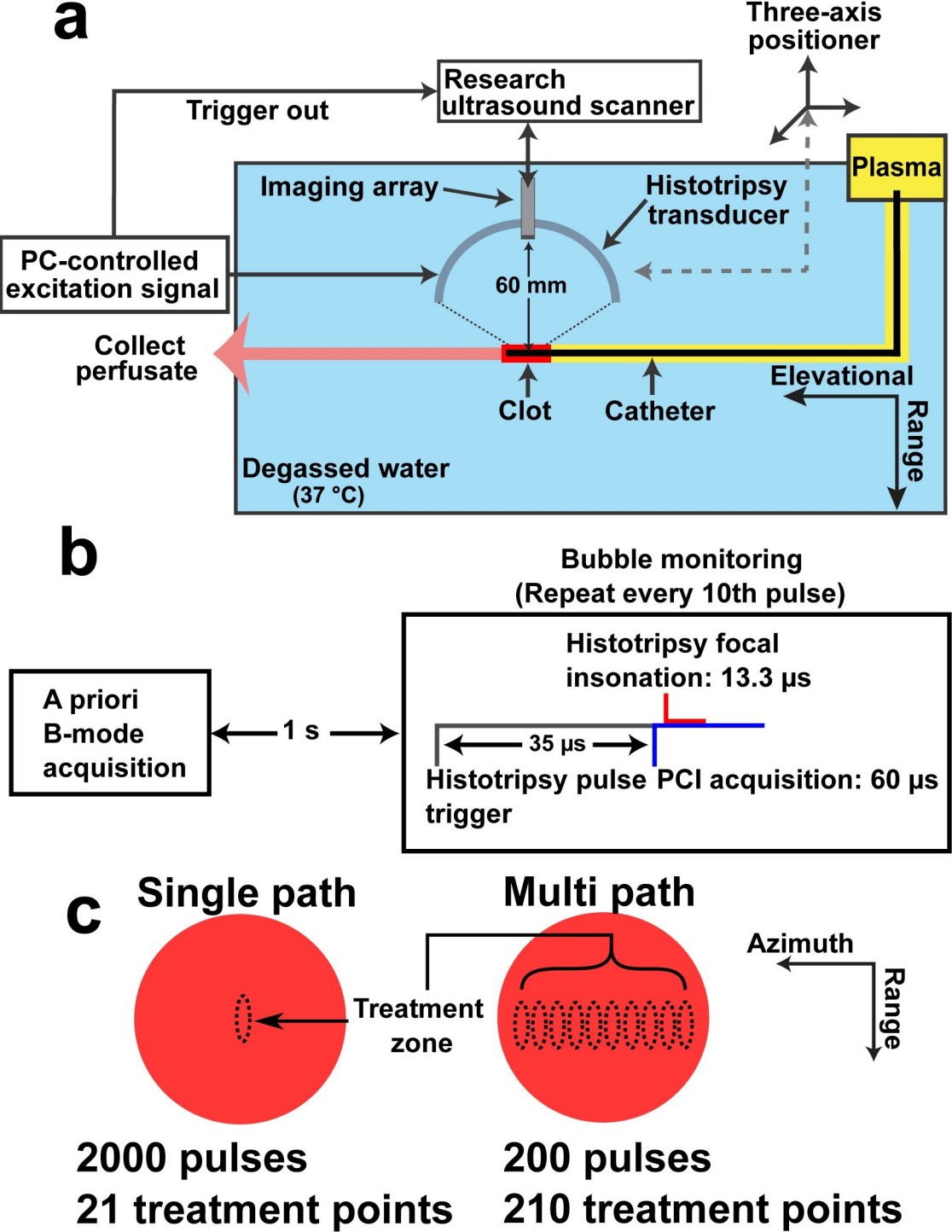

**Fig 1. Experimental setup.** (a) Overview of flow channel system. Plasma flowed from right to left in the figure. Bubble activity was initiated in the clot with a focused histotripsy source. Lytic was infused directly within the clot via the multi-side port catheter. The 'Elevational' and 'Range' dimensions correspond to the imaging plane for the diagnostic imaging array. The 'Azimuthal' dimension of the imaging plane (i.e. along the elements of the array) is into the page. (b) Timing diagram for the acquisition of acoustic emissions generated during histotripsy insonation. Passive acquisition of acoustic emissions was initiated 5 µs before the histotripsy pulse reached the focus. (c) Cross sectional view of clot with labeled insonation locations for single and multi-path histotripsy exposure schemes. For the single-path exposure scheme, bubble activity was generated at 21 discrete locations along the length of the clot (i.e. 21 different cross sections of the clot), and the focus targeted the center of the clot cross section (dashed line). The multi-path scheme scanned the histotripsy source along the length of the clot ten times, with each scan displaced by 0.25 mm laterally to the

central axis of the clot. In this figure, the central axis of the clot is directed into the page. There were a total of 210 total treatment points for the multi-path scheme (21 treatment points per scan). A total of 2,000 pulses were applied per treatment location for the single-path scheme, and 200 pulses for the multi-path approach. The total treatment time was ~ 21 min for both schemes. The 'Range' and 'Azimuthal' dimensions correspond to the imaging plane for the diagnostic imaging array. The 'Elevation' dimension of the imaging plane (i.e. perpendicular to the imaging plane) is into the page.

patients [39]. The flow channel was submerged in a tank of degassed ($<$ 20% dissolved oxygen), reverse osmosis water heated to 37.3˚C ± 0.5 ˚C. To be consistent with current treatment strategies for DVT [6], rt-PA was administered via a multi-side port, 5F catheter (Cragg-McNamara, Medtronic, Minneapolis, Minnesota, USA). The catheter was introduced into the model vessel via a hemostasis valve and positioned approximately through the central axis of the clot. No guidewires or sheaths were used to advance the catheter through the clot.

Pre-treatment planning was performed as described previously [32, 33]. Briefly, three orthogonal motorized linear stages (BiSlide, Velmex Inc., Bloomfield, NY, USA) were used to position the histotripsy source. Motor positions that aligned the histotripsy focus within the clot were determined in 5 mm increments along the length of the clot (3 in total). A path was then interpolated between these locations to assign discrete treatment points within the clot for histotripsy exposure. Two schemes were tested for histotripsy administration: single-path and multi-path (Fig 1c). For the single path method, treatment points with 0.5 mm spacing were assigned along the length of the clot (21 total insonation locations along the 'Elevational' dimension indicated in Fig 1a). Bubble activity was targeted to the center of the clot cross section. At each treatment location, 2,000 pulses of 13.3 μs pulse duration (nominally 20 cycles of the 1.5 MHz fundamental driving frequency) were applied at a 40 Hz rate (0.05% duty cycle). The focal pressure was calibrated for driving levels up to ~25 MPa. Bubble formation prevented calibration for larger driving levels, so the peak negative pressure level was estimated by linear extrapolation based on the available calibration as described previously [40]. Time-averaged intensities for histotripsy are typically $<$ 100 W/cm$^2$ because of the low duty cycle [25]. A prior study demonstrated these histotripsy insonation parameters consistently nucleate a bubble cloud and promote clot dissolution [33].

The multi-path scheme also scanned the histotripsy source along the length of the clot, but repeated the process ten times. For each scan, the histotripsy source was shifted 0.25 mm laterally relative to the central axis of the clot (along azimuthal direction noted in Fig 1c). A total of 210 treatment points were applied for the multi-path insonation scheme (10 scans along the length of the clot, 21 treatment points per scan). The number of histotripsy pulses applied per treatment location was reduced by a factor of ten for the multi-path method (200 pulses/location) to keep the treatment duration consistent between the single-path scheme (~ 21 min for both the single-path and multi-path schemes). Otherwise, the histotripsy exposure conditions were consistent between the two insonation schemes (35 MPa peak negative pressure, 20 cycle/13.3 μs pulse duration, 40 Hz pulse repetition frequency).

For the single-path insonation scheme, rt-PA was diluted into physiologic saline to a concentration of 0, 0.23, 2.25, or 4.54 μg/mL. An infusion pump (Masterflex 200, Cole Parmer, Vernon Hills, IL) was used to deliver the lytic solution through the infusion catheter directly into the clot at a fixed rate of 230 μL/min (corresponding rt-PA infusion rates of 0, 0.02, 0.20, or 0.41 mg/hr for each respective dilution). The highest rt-PA concentration is slightly lower than the the DVT standard of care (0.5 mg/hr for catheter-direct thrombolysis), and the second-highest dosage models pharmacomechanical therapy [8, 41]. The multi-path insonation scheme was tested with only the highest lytic dose (e.g. dose used as the DVT standard of care). Eight clots were tested for each arm, with a total of 72 clots used in the study.

Following treatment, the residual clot was removed from the flow model and weighed with a digital balance. The percent change in clot mass relative to that measured prior to treatment was used as the primary metric of therapeutic efficacy. Preliminary studies were also conducted to gauge damage to the catheter following histotripsy exposure. The instrument was examined with brightfield microscopy at 10x magnification. Observations indicated no mechanical damage to the catheter following exposure to histotripsy bubble activity.

## Perfusate assays

Following treatment, samples of the plasma perfusate were assayed for markers of clot degradation. Hemoglobin and D-dimer concentrations were measured as described previously to assess hemolysis and fibrinolysis, respectively [33]. Briefly, hemoglobin was measured via Drabkin's reagent assay (Sigma-Aldrich, St. Louis, MO, USA). Aminocaproic acid (100 μg/mL) was added to aliquots used to assess fibrinolysis directly after treatment in order to halt rt-PA activity. A latex immune-turbidimetric assay (STA R Max, Stago, Ansières sur seine, France) was used to quantify D-dimer in fibrinogen equivalent units. An assay (Diagnostica Stago Inc., USA) was used to determine total fibrin degradation products, including those non-specific to rt-PA activity. The assay for non-specific fibrin degradation products was semi-quantitative, and a range of concentrations were reported: 0–10 μg/mL, 10–20 μg/mL, or 20–40 μg/mL. Separate aliquots of the perfusate were assayed to determine the concentration of intact red blood cells and intact platelets shed from the clot during treatment using a hematology analyzer (XN-350, Sysmex, Lincolnshire, IL, USA).

## Tracking bubble activity

At each treatment location, the location and strength of bubble cloud activity generated during the histotripsy insonation was tracked with passive cavitation imaging (Fig 1b) [42]. Acoustic emissions generated by the bubble cloud during histotripsy exposure were passively recorded with the coaxial imaging array and processed offline using the robust Capon beamformer [43]. The pixel intensity of the passive cavitation image is proportional to the acoustic intensity emitted by the bubble cloud at that location (Fig 2). Due to data transfer rate limitations, acoustic emissions were recorded once for every ten histotripsy pulses generated (200 frames for the single-path histotripsy exposure scheme and 20 frames for the multi-path schemes). At each treatment location, the individual frames were summed pixel-wise to generate a cumulative image. The distance between the location of maximum acoustic emissions and infusion catheter (located with a B-mode ultrasound image co-registered with the passive cavitation image, Fig 2b) was tabulated for each treatment arm. The strength of emissions generated within the clot were also tabulated, and differences in bubble activity between arms were compared.

## Histology

After treatment, a central portion of the clot of ~ 3 mm thickness was fixed in 10% formalin for 24 hours, paraffin embedded, and sectioned to 5 μm thickness. Hematoxylin & Eosin staining was used to assess differences in the clot between treatment schemes. The slides were digitally scanned using 20x magnification and were reviewed by a board-certified pathologist.

## Statistical methods

Statistical analysis was performed using MATLAB 2017b (The Mathworks, Natick, MA, USA). Differences in the mass loss, hemoglobin concentration, D-dimer concentration, platelet

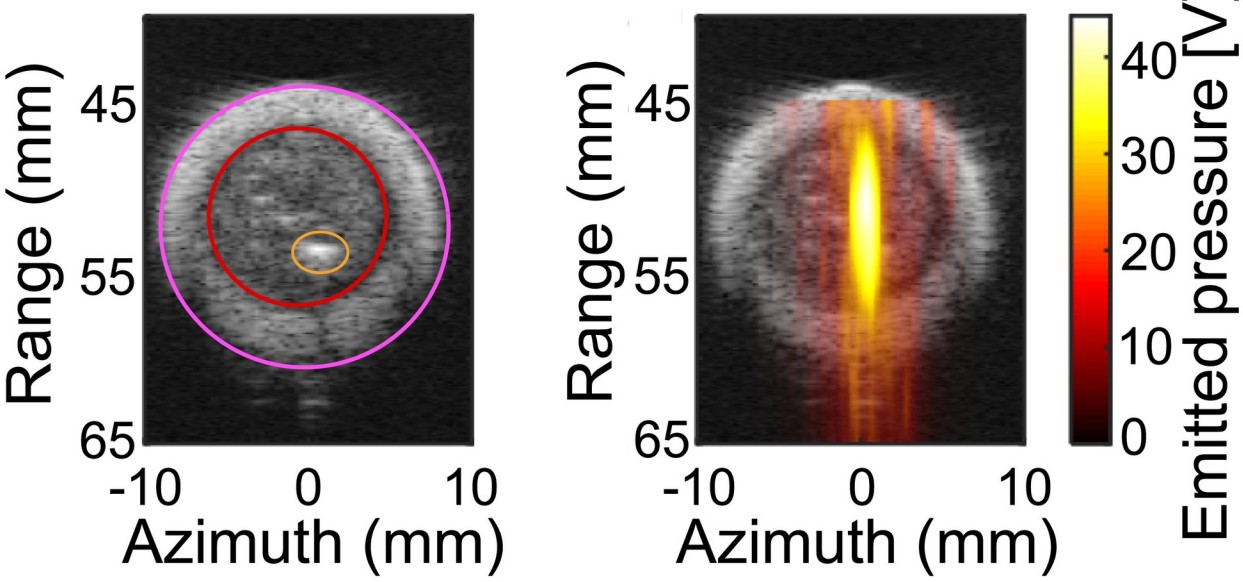

**Fig 2.** (Left) Segmented B-mode indicating the lumen (pink red line), clot (red line), and catheter (orange line). (Right) Duplex passive cavitation (hot colormap) and B-mode image. The histotripsy pulse propagates from top to bottom in this image. The pixel intensity is proportional to the acoustic pressure generated by the bubble cloud at that location, reported here as the voltage generated by the incident pressure wave on the elements of the array.

count, and red blood cell count between arms were assessed using a Wilcoxon ranked-sum test with Bonferroni correction for multiple comparisons [44, 45]. Clot degradation components were compared to mass loss using multiple linear regression. The coefficients of the regression were reported as the confidence intervals of β values. All statistical tests used a significance level of 0.05.

## Results

### Overall treatment efficacy

The clot mass loss for all treatment arms is reported in Fig 3. Histotripsy alone generated no significant increase in mass loss compared the control arm (no lytic and no histotripsy). The single-path histotripsy scheme increased clot mass loss compared to lytic alone for the two lowest lytic doses (0.23 μg/mL and 2.25 μg/mL). No arms using the single-path histotripsy insonation scheme increased mass loss relative to lytic alone at 4.54 μg/mL, the rt-PA dosage used in catheter-directed-thrombolysis. The multi-path scheme increased clot mass loss relative to lytic alone, but not relative to the single-path scheme (4.54 μg/mL rt-PA dosage only).

### Assessment of cellular clot debris following histotripsy exposure

Hemoglobin, intact erythrocytes, and platelets in the plasma perfusate following treatment are reported for all arms in Fig 4. In the absence of rt-PA, histotripsy increased hemoglobin production relative to control (no histotripsy). For arms with rt-PA, there was no difference in the hemoglobin concentration in the plasma for arms with or without histotripsy exposure (single- or multi-path histotripsy insonation schemes). For a given lytic concentration, no differences were observed in the concentration of red blood cells for arms with or without histotripsy (single- and multi-path insonation scheme). Similarly, neither histotripsy insonation scheme (single or multi-path) had an influence on platelets shed from the clot.

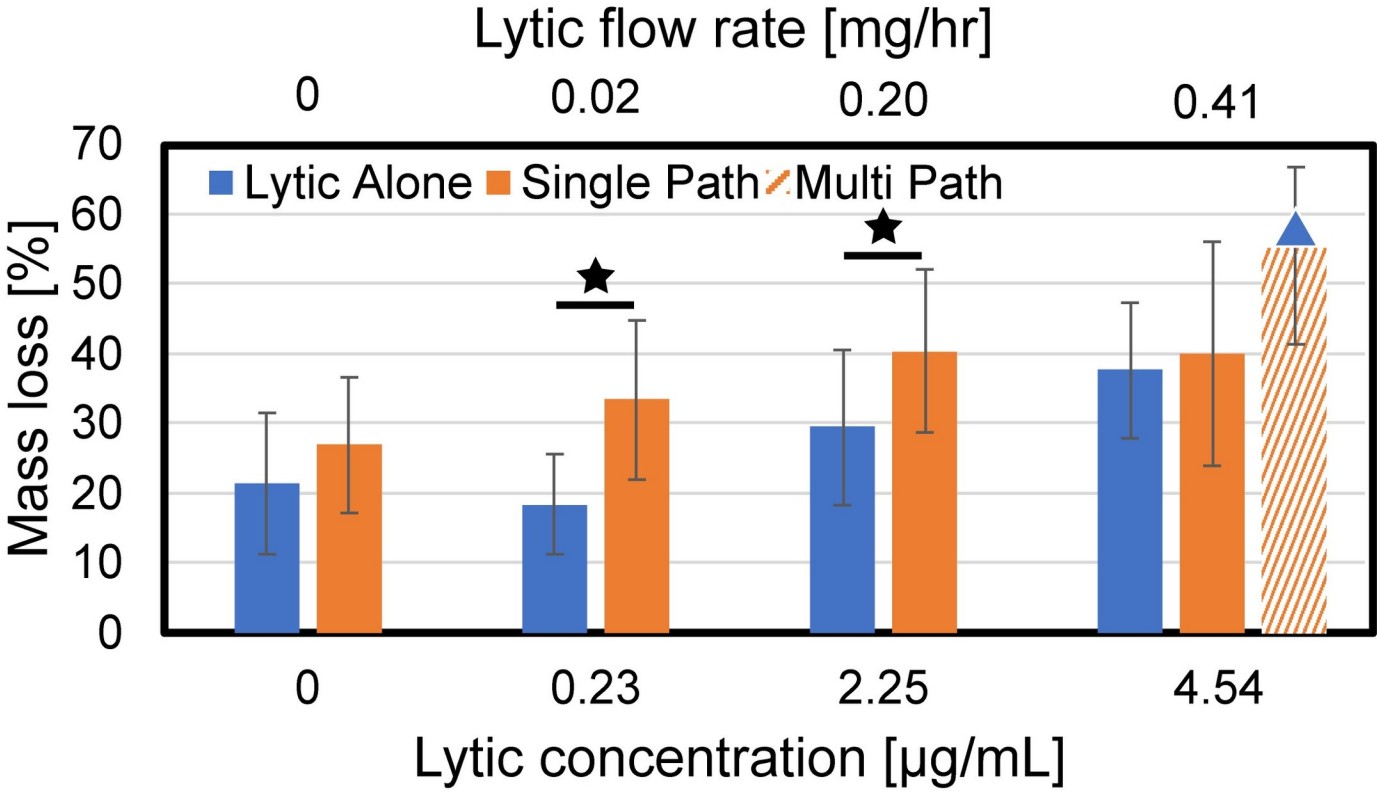

**Fig 3. Clot mass loss for the treatment arms tested in this study as a function of the lytic rt-PA concentration.** The legend indicates the insonation scheme (histotripsy was not applied for the 'Lytic Alone' arm). Stars indicate statistically significant increases in mass loss for a given rt-PA dose (p < 0.05). The blue triangle indicates an increased mass loss compared to the standard of care arm (4.54 µg/mL rt-PA only).

### Assessment of fibrinolysis following histotripsy exposure

The generation of D-dimer is reported in Fig 5 for each treatment arm. No appreciable D-dimer was generated via histotripsy alone compared to control samples (no histotripsy, no lytic). Lytic doses 0.23 µg/mL and greater combined with histotripsy resulted in D-dimer levels equivalent to that for rt-PA alone at 4.54 µg/mL (standard-of-care rt-PA dosage for catheter-directed thrombolysis). For clots exposed to 2.25 µg/mL of rt-PA, histotripsy significantly increased D-dimer relative to lytic alone. The multi-path insonation scheme increased D-dimer compared to lytic alone, but not relative to the single-path insonation scheme (4.54 µg/mL rt-PA dosage only).

The concentration of non-specific fibrin degradation products (FDPs) generated for each treatment arm are reported in Fig 6. For rt-PA doses of 0 and 0.23 µg/mL, no differences were noted in FDP production for arms with or without histotripsy. For the rt-PA doses that model pharmacomechanical therapies (2.25 µg/mL) or catheter-directed thrombolysis (4.54 µg/mL), higher concentrations of FDPs were more frequently observed for histotripsy arms relative to administration of lytic alone. Similarly, the multi-path scheme skewed FDP production to higher concentrations compared to lytic alone. No appreciable differences were noted for FDPs generated by the multi-path histotripsy scheme relative to the single-path histotripsy approach.

### Linear regression between debris components and mass loss

Multiple linear regression analysis was used to determine the relationship between mass loss and clot debris generated for each arm. Only the single-path insonation scheme was included

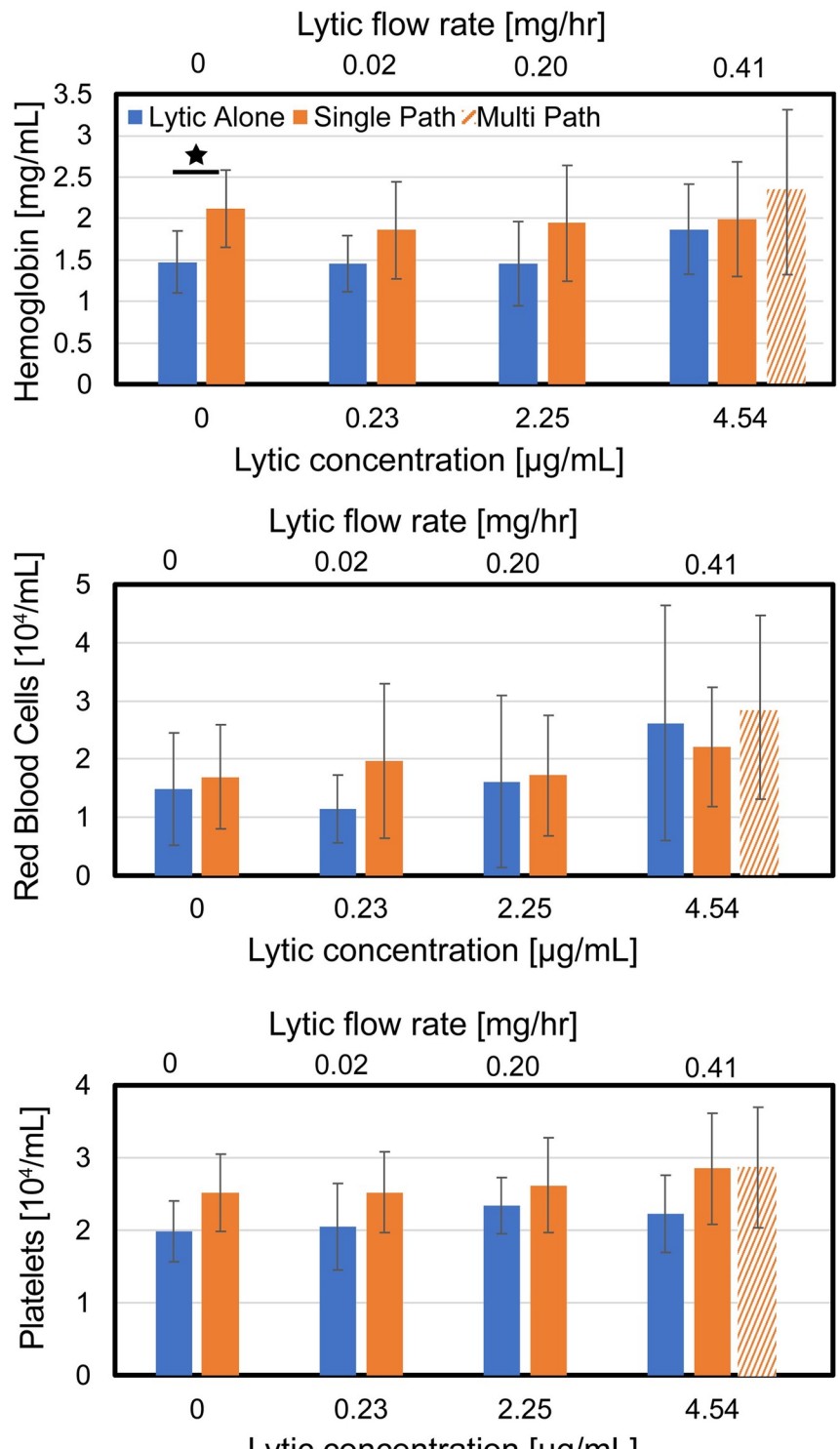

**Fig 4. Analysis of cellular clot debris in the perfusate following treatment as a function of the lytic rt-PA concentration.** (Top panel) Hemolysis, indicative of red blood cell lysis, (Middle panel) Intact red blood cells, and (Bottom panel) Intact platelets. Stars indicate a statistically significant difference ($p < 0.05$). The legend in the top panel indicates the insonation scheme, and is consistent for all panels. Histotripsy was not applied for the 'Lytic Alone' arm.

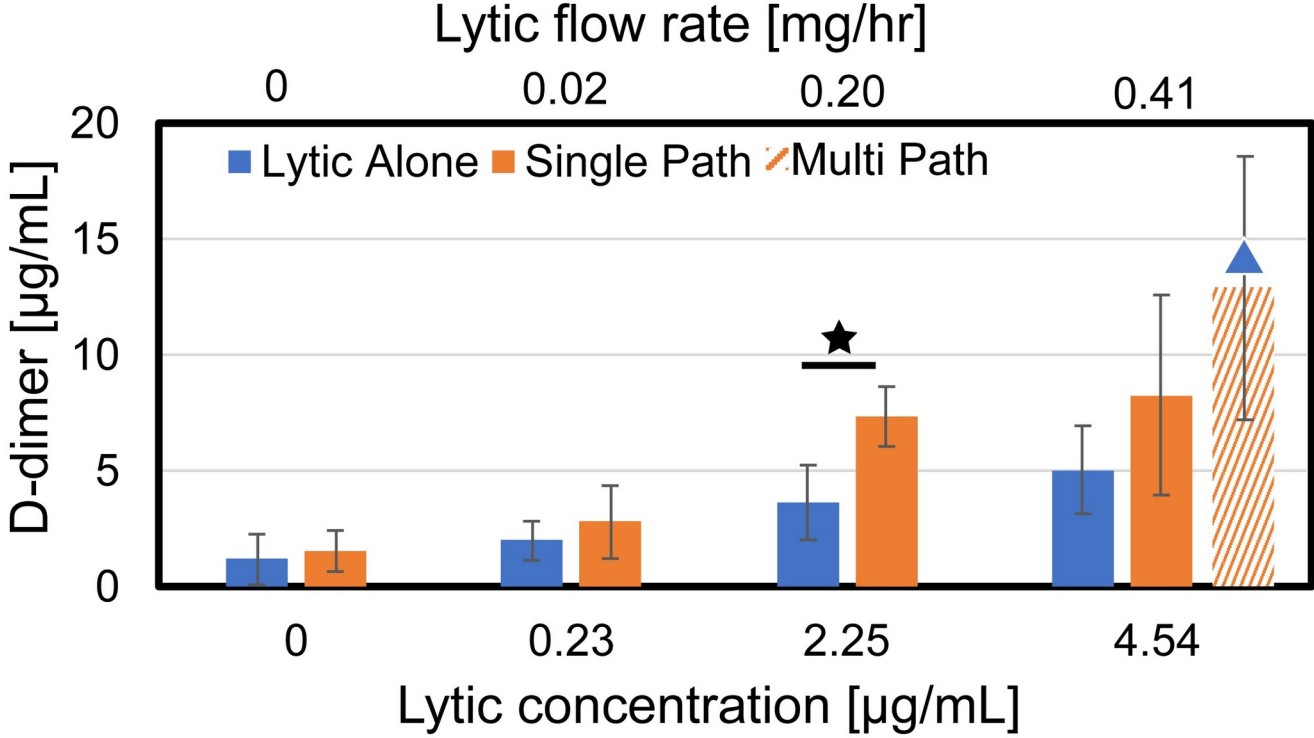

**Fig 5. The concentration of D-dimer generated for each treatment arm.** Stars indicate statistically significant increases in mass loss for a given concentration of the lytic rt-PA ($p < 0.05$). The blue triangle indicates a significant increase compared to the standard of care arm (4.54 μg/mL, no histotripsy). The legend indicates the insonation scheme used for clot exposure.

in this analysis. The 95% confidence intervals for the slopes of the linear regression are presented in Table 1. As a result of the normalization process these values have units of percent mass loss (%) relative to unit increase in concentration for each component. Clots exposed to lytic alone exhibited a significant relationship between mass loss, hemoglobin, and D-dimer. For clots exposed to rt-PA and histotripsy, mass loss exhibited a significant relationship with hemoglobin, D-dimer, and intact erythrocytes.

### Location and strength of bubble cloud activity and assessment of catheter

The average Euclidean distance between the maxima of acoustic emissions and catheter location for all histotripsy arms was 3.05 ± 2.69 mm (single-path insonation scheme only). No differences were noted between treatment arms and the position of bubble cloud activity. Similarly, there were no differences in the strength of emissions generated between treatment arms.

### Structural clot damage assessed via histology

Representative clot samples following exposure to histotripsy and/or lytic are shown in Fig 7. Control samples (no histotripsy, lytic, or interaction with the infusion catheter) had a fibrin-rich rim of ~ 1 mm thickness. The interior of the clot was erythrocyte-rich, with a reduced prevalence of the dense fibrin structure observed at the exterior. For treated clots (with or without histotripsy), damage was observed adjacent to the catheter location. The extent of damage was larger than the catheter (~ 1.67 mm diameter), suggesting the sample was torn by passage of the catheter. Red blood cell fragmentation and fibrin reduction near the catheter

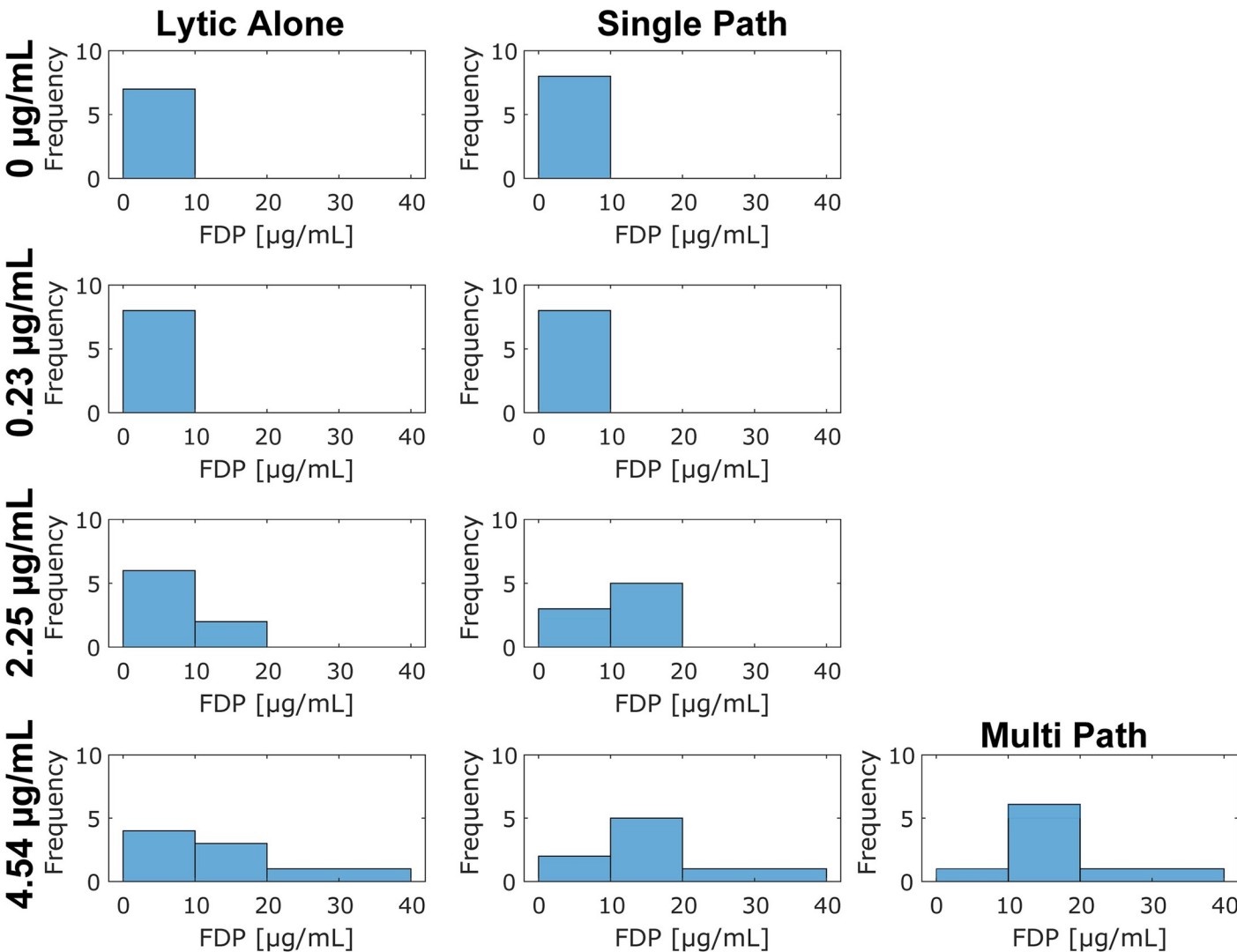

**Fig 6. Observed concentration of non-specific fibrin degradation products (FDPs) generated for each treatment arm.** The concentration of rt-PA is noted along the left. The left column of subfigures corresponds to rt-PA-only arms, and the right column of subfigures corresponds to histotripsy and lytic. The multi-path histotripsy exposure scheme is labeled in the third column. The assay is semi-quantitative, so concentrations are reported in bins from 0 to 10 μg/mL, 10 to 20 μg/mL, and 20 to 40 μg/mL. The number of datasets observed within each bin are reported for each arm, as indicated by the 'Frequency' on the subplot ordinate.

tract were noted, but there were no consistent observations to differentiate between arms with and without histotripsy.

## Discussion

### Catheter/Bubble cloud interactions

The use of histotripsy to improve drug bioavailability, including treatment with a thrombolytic agent, has been an active area of research [29, 31, 32]. To be consistent with the standard of practice for the treatment of acute proximal venous thromboembolism [6], rt-PA was adminis-tered in this study directly in the clot via an infusion catheter. A concern was that ports in the catheter may serve as prompt nucleation sites that reduce the pressure threshold necessary for bubble generation and diminish treatment efficacy [46]. Passive imaging indicated bubble

**Table 1. The slope tabulated with multiple linear regression analysis between mass loss and clot components generated due to exposure to rt-PA alone, or rt-PA and histotripsy.**

| Clot component | Effect of clot component on mass loss [%] | |
| --- | --- | --- |
| | rt-PA Alone | Histotripsy and rt-PA |
| Hemoglobin | (3.1, 11.0) | (3.3, 9.0) |
| D-dimer | (1.2, 6.7) | (4.0, 8.0) |
| Red blood cell | (-0.7, 6.0) | (1.8, 6.5) |
| Platelet | (-3.9, 1.9) | (-4.0, 1.4) |

Slopes are reported as 95% confidence intervals. As a result of the normalization process these values have units of percent mass loss (%) relative to unit increase in concentration for each component. The middle column represents regression slopes for clots exposed to rt-PA alone, and the right column represents slopes for clots exposed to rt-PA and the single-path histotripsy insonation scheme.

clouds were not preferentially nucleated on the surface of the catheter, possibly because of the highly focused source used here [38]. An additional finding was that histotripsy generated no mechanical damage to the catheter. Histotripsy is effective in breaking up soft tissues, but its ablative qualities decrease with the tissue stiffness [28, 47]. The catheter is comprised of a

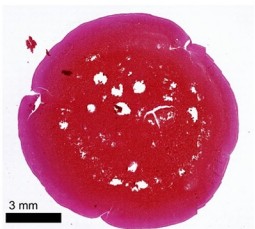

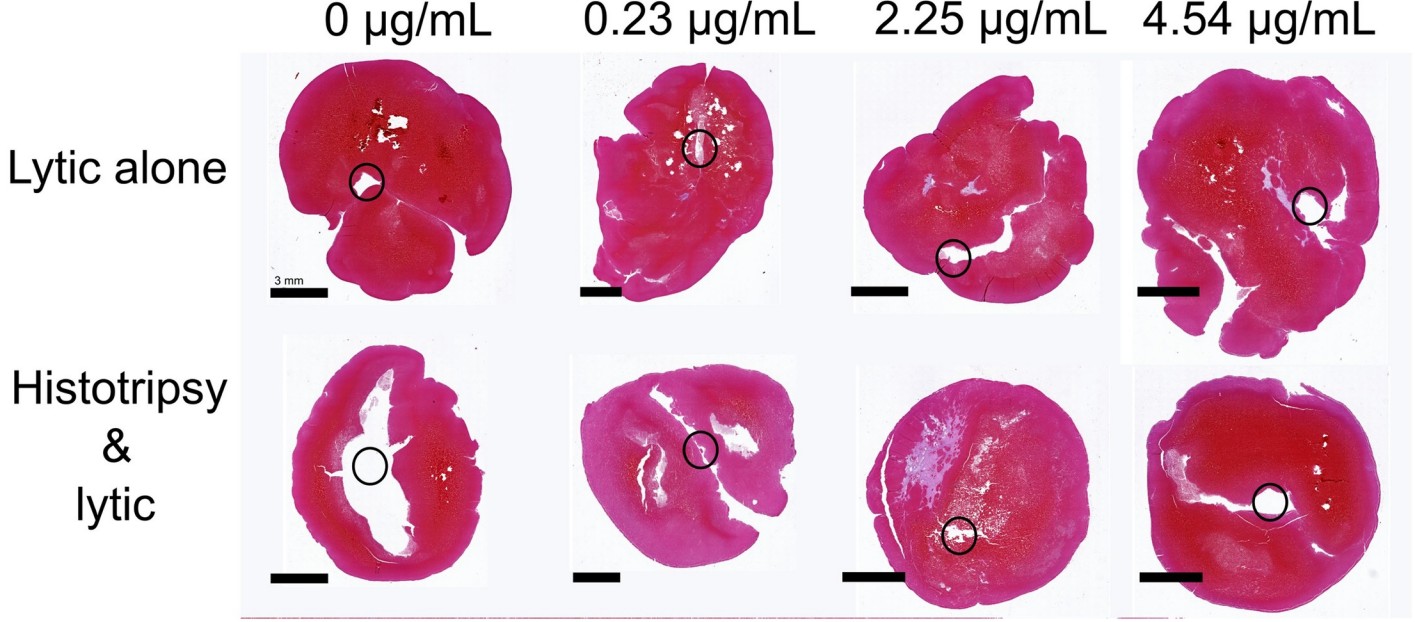

**Fig 7. Representative clots for each rt-PA dose explored for the single-path histotripsy treatment arms.** The scale bar corresponds to a 3 mm distance. The top image represents a control sample (no histotripsy, no lytic, and no catheter). The gaps within this control clot are histological artifacts. The concentration of rt-PA infused is noted at the top of each column for the bottom two rows (sections for samples treated with lytic and histotripsy). Black circles depict the approximate location of the catheter.

plastic material with a larger elastic modulus than soft tissues [48], and therefore appears to be relatively impervious to damage from histotripsy exposure. Histotripsy bubble activity must act concurrently with rt-PA administration to improve fibrinolysis [33]. The data collected here indicate that histotripsy bubble clouds can be initiated in conjunction with CDT to improve rt-PA activity, and therefore treatment outcomes.

## Histotripsy treatment efficacy

Prior studies have demonstrated that combining histotripsy and rt-PA cause clot degradation more effectively than either treatment scheme individually at doses currently used with venous pharmacomechanical approaches [33, 32]. The data in this study indicate fibrinolysis is equivalent between the standard catheter-directed thrombolysis dosing scheme and a twenty-fold reduction in rt-PA when combined with histotripsy (Fig 5). A reduction in lytic dose may reduce the likelihood of bleeding associated with rt-PA administration [9, 34]. Risks associated with histotripsy may include off-target damage, though these effects can be minimized using appropriate sources and pulsing schemes [38]. Histotripsy exposure may also result in regional endothelial denudation [15], consistent with the damage profile of current clinical thrombectomy devices [49]. It should be noted that restoration of flow is the clinical endpoint, whereas mass loss and D-dimer were used in this study to assess treatment efficacy. Vessel patency during thrombolysis has been shown to be correlated with D-dimer [50], providing some validity to the generalizability to these findings. Future studies should incorporate flow as a metric of thrombolytic efficacy and treatment outcome [51].

In the absence of lytic, histotripsy did not increase mass loss relative to control (Fig 3). A number of previous studies indicate effective clot degradation from histotripsy exposure alone [31, 32, 52]. There are multiple potential reasons for the lack of increased clot degradation from histotripsy exposure in this study. The infusion catheter imparted direct damage to the clot as indicated with histological observation (Fig 7). Catheter damage resulted in an increased baseline mass loss compared to prior studies. For instance, control samples (no histotripsy and no rt-PA) in this study had a mass loss of ~ 20%, nearly double that observed in prior studies [32, 33]. The clots tested here were larger compared to prior investigations (10 mm diameter vs. 4 mm). The reduced fraction of clot volume exposed to bubble activity may be insufficient to produce notable outcomes for the single-path histotripsy exposure scheme. The clot size may also have affected the bubble dynamics. The multi-cycle histotripsy pulse used in this study nucleates bubbles via a scattering mechanism [53], which requires a shocked profile for the incident pressure waveform. The histotripsy pulse will be attenuated as it propagates through the clot, particularly for the high frequency components necessary for shockwave formation [54]. Hence, robust bubble cloud formation via shock scattering may be diminished for the clot model tested here.

Mass loss and D-dimer production were enhanced for the single-path histotripsy arms relative to lytic alone for the two lowest doses of rt-PA (0.23 and 2.25 μg/mL), but not the highest lytic dose (Figs 3 and 5). The lack of histotripsy enhancement for the highest lytic dose may be a reflection of the clot model used here. The fibrin network was sparse towards the center of the clot (Fig 7). Fibrinolysis may have been saturated in close proximity to the catheter (i.e. near the center of the clot) for the highest rt-PA dose. In contrast, the rim of the clot was fibrin-rich, possible due to interaction between platelets and the borosilicate glass while the clot was forming [55]. Fibrin meshes formed with activated platelets are less susceptible to cleavage by rt-PA [56]. Hence, the fibrin-rich outer portions of the clot may benefit the most from enhanced lytic delivery. While targeted to the center of the clot for the single-path insonation scheme, bubble activity was targeted throughout the clot for the multi-path histotripsy

scheme, including the fibrin-rich rim. Such a scheme may therefore be necessary to ensure enhanced delivery can be achieved in the portions where enhanced delivery is critical (i.e. near the periphery of the clot model used here).

Improved fibrinolysis and mass loss were noted with the multi-path histotripsy treatment scheme relative to the lytic alone (rt-PA dose 4.54 μg/mL). Microchannels may be generated during early passages along the clot which increase surface area exposed to rt-PA and thus the rate of thrombolysis [57]. A prior study noted the importance of generating bubble activity in close proximity to the lytic for improved thrombolytic efficacy [33]. In this study, the catheter position did not always overlap with the histotripsy focal zone for the single-path insonation scheme. The multi-path histotripsy scheme ensured bubble activity was generated in close proximity to the infusion location at some point during treatment. Note that the multi-path exposure scheme was not examined in the absence of lytic. Hemoglobin is the primary clot degradation product generated for application of histotripsy alone [33]. The multi-path method did not increase hemoglobin production relative to the single-path scheme (Fig 4), suggesting a similar degree of mechanical ablation between the two exposure protocols. Future studies will investigate the influence of insonation scheme on clot hemolysis in more detail.

The number of pulses per treatment location was reduced for the multi-path histotripsy scheme to ensure the overall treatment time and number of applied pulses was the same as the single-path scheme (200 pulses per treatment point vs. 2,000 pulses, respectively). Mass loss was equivalent between the two insonation schemes, suggesting there may be no benefit to apply more than 200 pulses per location. This finding is consistent with a prior study that noted limited changes in the ablated volume when more than 500 histotripsy pulses were applied under similar insonation condition [58]. Catheter-directed thrombolytics are administered over the course of several days [9], much longer than the treatment time considered here (~ 21 min). We anticipate it is possible to optimize the histotripsy insonation scheme and reduce the treatment duration, minimizing the potential for off-target effects. Even for the unoptimized scheme tested here, the results in this study indicate histotripsy could significantly reduce the overall duration required for rt-PA administration. Further studies are needed to determine appropriate dosing schemes to gauge the point of appropriate histotripsy pulsing strategies that are effective and efficient in the promotion of hemolysis and fibrinolysis.

### Histotripsy and mechanisms of clot disintegration

It is hypothesized that histotripsy in combination with rt-PA promotes clot dissolution through two primary mechanisms: hemolysis and enhancement of fibrinolysis. No differences were noted in the generation of hemoglobin for arms with and without histotripsy in the presence of lytic (Fig 4), in contradiction to a prior study [33]. The discrepancy between these findings may be attributed to the interaction between the catheter and clot used in this study. Hemoglobin production for control arms (no histotripsy, no lytic) was increased here compared to the prior experiment (~ 1.5 mg/mL for current study vs. ~ 0 mg/mL [33]). These results indicate the degree of hemolysis generated by histotripsy is within the natural variation caused by interactions between the clot and infusion catheter. The muted hemolysis caused by histotripsy may be a clinical boon. Free hemoglobin causes nitric oxide scavenging and vascular dysfunction [59]. The similarity of hemolysis for arms with and without histotripsy suggest that endothelial dysfunction would not be a significant procedural complication, though further *in vivo* tests are need for confirmation.

Fibrin is cleaved by plasmin at discrete locations during thrombolysis. Cross-linked portions of the fibrin mesh can be cleaved into variable fragments from 260 kDa (D-dimers) to

greater than 2300 kDa (macro-fragments) [60]. Histotripsy alone generated no detectable fibrin degradation products relative to control arms (Figs 5 and 6), consistent with prior studies indicating minimal breakdown of stiff, extracellular structures with histotripsy [28]. The single-path insonation scheme improved fibrinolysis (i.e. D-dimer release) for the 2.25 μg/mL dose of rt-PA, though not for the lowest and highest lytic doses (0.23 and 4.54 μg/mL, respectively). In contrast, the release of non-specific fibrin degradation products was increased for histotripsy arms at the two highest lytic doses (Fig 6). The lack of increased D-dimer production with an increased generation of non-specific fibrin degradation products may indicate histotripsy improves delivery of the lytic to the fibrin mesh for these arms, but insufficient fibrinolytic activity occurs to break the mesh down fully to the minimal D-dimer unit.

Intact red blood cells and platelets shed from the clot during treatment in the perfusate were also tracked in this study (Fig 4 and Table 1). For all treatment arms, clot mass loss was correlated with hemoglobin and D-dimer production. The mass loss for histotripsy arms also correlated with intact erythrocytes (Table 1), though no differences were noted in the concentration of erythrocytes or platelets for arms with or without histotripsy (Fig 4). The lack of difference in erythrocytes for treatment arms with or without histotripsy may be attributed in part to disruption of the liberated cells under bubble cloud activity. Prior studies have noted liberated bubble-induced vortices can trap particulates, eroding them to acellular debris over time [30]. Further analysis on the liberation of cellular components under histotripsy exposure is warranted.

### Limitations

There are limitations to this study that restrict the generalizability of these findings. The flow rate of plasma around the clot was held constant, whereas changes in blood flow would likely occur over the course of recanalization which would contribute to the thrombolytic profile [61]. Damage to the clot due to the catheter may contribute to mass loss and perfusate debris. Because of embolization risk during CDT, interventionalists may place a filter in the inferior vena cava [62]. The filter may alter the flow profile of venous return, whereas here a fixed flow rate was used. The fibrin degradation assay used here was semi-quantitative, which could be insensitive to small changes in the breakdown of fibrin due to the treatment schemes tested here. The *in vitro* clots used in this study were homogenous, which may not be representative of heterogeneous pathologies *in vivo* [10]. Here, studies were conducted in a tank of degassed water to couple the acoustic field to the clot. A bath of degassed water with an acoustic window will be required to target the clot *in vivo*, and the ultrasound pulse will have to transverse several centimeters of tissue [15, 63]. Three orthogonal motorized stages were used to translate the histotripsy source, which is effective for the controlled setup in this *in vitro* study. Additional flexibility for transducer placement will be required for treatment *in vivo*. One possible solution to distribute bubble activity throughout the clot burden are collaborative robots, which have six degrees-of-freedom to enable arbitrary angulation of the histotripsy source relative to the target [64]. Finally, physiological effects due to interaction between the clot and the venous wall were not replicated by this *in vitro* model.

### Conclusions

These studies indicate that the addition of histotripsy to catheter-directed thrombolysis for venous thrombosis may allow for a reduction in rt-PA dose. Data collected here indicate that, in the presence of histotripsy, the dose of catheter-directed thrombolytics can be reduced by a factor of twenty relative to the standard dose and retain equivalent clot mass loss and fibrinolysis in an *in vitro* model. Although histotripsy did not remove the fibrin mesh on its own, it

significantly enhanced thrombolytic activity for the lytic doses examined in this study. Furthermore, catheter placement did not impede our ability to position the histotripsy bubble cloud. These results indicate that histotripsy can serve as an effective adjuvant therapy for catheter-directed thrombolysis.

## Acknowledgments

The authors would like to thank the University of Chicago medical labs, the University of Chicago Biophysics Core Facility, and the University of Chicago Human Tissue Resource Facility. Discussions with Dr. Adam Maxwell at the University of Washington and Drs. Kevin Haworth and Chadi Zemzemi at the University of Cincinnati were also helpful in developing these studies.

## Author Contributions

**Conceptualization:** Samuel A. Hendley, Christy K. Holland, Jonathan D. Paul, Kenneth B. Bader.

**Data curation:** Samuel A. Hendley, Osman Ahmed, Jonathan D. Paul, Kenneth B. Bader.

**Formal analysis:** Samuel A. Hendley, Aarushi Bhargava, Geoffrey D. Wool.

**Funding acquisition:** Kenneth B. Bader.

**Investigation:** Samuel A. Hendley, Aarushi Bhargava.

**Methodology:** Samuel A. Hendley, Aarushi Bhargava, Osman Ahmed, Jonathan D. Paul, Kenneth B. Bader.

**Project administration:** Kenneth B. Bader.

**Resources:** Osman Ahmed, Jonathan D. Paul, Kenneth B. Bader.

**Software:** Samuel A. Hendley, Aarushi Bhargava.

**Supervision:** Kenneth B. Bader.

**Validation:** Samuel A. Hendley, Aarushi Bhargava, Geoffrey D. Wool.

**Visualization:** Samuel A. Hendley, Geoffrey D. Wool.

**Writing – original draft:** Samuel A. Hendley.

**Writing – review & editing:** Aarushi Bhargava, Christy K. Holland, Geoffrey D. Wool, Osman Ahmed, Jonathan D. Paul, Kenneth B. Bader.

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
