## [Decision Letter · Decision Letter 0]

18 Oct 2021

PONE-D-21-29418(More Than) Doubling Down: Effective fibrinolysis at a reduced rt-PA dose for catheter-directed thrombolysis combined with histotripsyPLOS ONE

Dear Dr. Bader,

Thank you for submitting your manuscript to PLOS ONE. After careful consideration, we feel that it has merit but does not fully meet PLOS ONE’s publication criteria as it currently stands. Therefore, we invite you to submit a revised version of the manuscript that addresses the points raised during the review process.

We look forward to receiving your revised manuscript.

Kind regards,

Virginie Papadopoulou, Ph.D.

Academic Editor

PLOS ONE

Journal Requirements:

"This work was funded by the National Heart, Lung, and Blood Institute (https://www.nhlbi.nih.gov/), Grant Number R01 HL133334"

"This work was funded in part by the National Institutes of Health under Grant R01 HL133334."

"This work was funded by the National Heart, Lung, and Blood Institute (https://www.nhlbi.nih.gov/), Grant Number R01 HL133334"

6. Thank you for stating the following in the Competing Interests section: 

"O. A. has acted as a consultant for Inari Medical, Boston Scientific, and received research grants from Inari Medical, Canon Medical, and Philips. He acted as a speaker and received compensation for Argon Medical, Canon Medical, Penumbra, Philips, and Johnson and Johnson. G.D.W. received honoraria and serves on the advisory board for Diagnostica Stago."

Additional Editor Comments:

Thank you for submitting your manuscript to PLOS ONE. This has now been reviewed by three external referees who have unanimously recommended a minor revision. Kindly consider their comments and questions below and provide a point by point reply, as well as revised manuscript during resubmission. 

Reviewers' comments:

Reviewer's Responses to Questions

**Comments to the Author**

1. Is the manuscript technically sound, and do the data support the conclusions?

Reviewer #1: Yes

Reviewer #2: Yes

Reviewer #3: Yes

2. Has the statistical analysis been performed appropriately and rigorously? 

Reviewer #1: Yes

Reviewer #2: Yes

Reviewer #3: Yes

3. Have the authors made all data underlying the findings in their manuscript fully available?

Reviewer #1: Yes

Reviewer #2: Yes

Reviewer #3: Yes

4. Is the manuscript presented in an intelligible fashion and written in standard English?

Reviewer #1: Yes

Reviewer #2: Yes

Reviewer #3: Yes

5. Review Comments to the Author

Reviewer #1: This paper reports an interesting sonothrombolysis by using catheter infusion and histotripsy. The findings on reduced tPA is promising. The following comments may be addressed to strengthen the manuscript:

1. Abstract

Page 2. Line 29. “… rt-PA at a one-twentieth dose reduction (0.23 μg/mL) when combined with histotripsy.”. This should be changed to “… rt-PA at a reduction to one-twentieth of the common clinic dose (0.23 μg/mL) when combined with histotripsy.”.

Page 2. Line 36. “Overall, the data collected in this study indicate the rt-PA dose can be reduce by…”. This should be changed to “Overall, the data collected in this study indicate the rt-PA dose can be reduced by…”. (use past tense for “reduce”)

2. Introduction

Page 3. Line 49. ” This approach is not without…”. This could be changed to “This approach has…” to be more concise.

Page 3 Line 55. Authors may add a paragraph here to introduce the status of sonothrombolysis including the ultrasound catheter based sonothrombolysis (e.g. EKOS, forward-viewing catheter based sonothrombolysis), focused ultrasound based sonothrombolysis, etc. , and their limitations, and then come to the uniqueness of the histotripsy based sonothrombolysis that is reported in this paper.

3. Methods and materials

Page 5 Line 74-81. Authors may comment on the clot retraction. Are they retracted clots or unretracted ones?

Page 6. Line 109. “Bubble activity was initiate in the clot…”. This should be changed to “Bubble activity was initiated in the clot…” so that “was” and “initiated” are both past tense.

Page 7. Line 129. “Motor positions that contained bubble activity within the clot were assigned in 5 mm increments along the length…”. I’m not sure if it is correct to say that a motor position can contain bubble activity. How about this: “Motor positions that correspond with bubble activity within the clot were assigned in 5 mm increments along the length…”.

Page 7. Line 139. “…scanned the histotripsy source scanned along the length…”. Remove the second “scanned” so that this phrase is written as “…scanned the histotripsy source along the length...”.

Page 8. Line 156. “A total of eight clots were tested for each arm, and a total of 72 total clots used in the study.”. This should be changed to: “Eight clots were tested for each arm, with a total of 72 clots used in the study.” to avoid repeating “total” 3 times.

Page 8. Line 158. “…the residual clot was removed from the flow model and was weighed…”. The second “was” can be removed to yield: “…the residual clot was removed from the flow model and weighed…”.

Page 8. Line 161. “The instrument was examination with brightfield microscopy…”. For this sentence, “examined” should be used. The sentence becomes: “The instrument was examined with brightfield microscopy…”.

4. Results

Page 11. Line 232. “Hemoglobin, intact erythrocytes, and intact platelets shed from the clot are reported for all treatment arms in”. This sentence/section needs to be completed.

Page 12. Line 241. There was already a “Figure 4” mentioned previously.

Page 12. Line 248. “Figure Error! No text of specified style in document.”. Add the figure number.

Page 12. Line 253. “…histotripsy significantly increased D imer…”. Change “D imer” to “D-dimer” so that the phrase becomes “…histotripsy significantly increased D-dimer…”.

Page 13. Line 257. “Figure Error! No text of specified style in document.:”. Add the figure number.

Page 13. Line 268. “The multi-path scheme was similarly skewed FDP production to higher concentrations compared to lytic alone.”. Change this to: “Similarly, the multi-path scheme skewed FDP production to higher concentrations compared to lytic alone.”.

Page 15. Line 299. “The average Euclidean distance between acoustic emissions maxima and catheter location…”. Change “emissions” to “emission’s” so that it is possessive. The phrase becomes: “The average Euclidean distance between the acoustic emission’s maxima and catheter location…”.

5. Discussion

Page 18. Line 357. “The histotripsy pulse will be attenuated as it propagated through the clots…”. Change “propagated” to “propagates” so that the tenses in the sentence correspond chronologically. The phrase becomes: “The histotripsy pulse will be attenuated as it propagates through the clots…”.

Page 20. Line 415. “…the release of non-specific fibrin degradation products was increased for histotripsy arms at two highest lytic dose…”. This should be changed to “…the release of non-specific fibrin degradation products was increased for histotripsy arms at the two highest lytic doses…” such that “dose” is switched to plural.

Page 21. Line 436. “The in vitro clots used in this study were homogenous, and which may not be representative…”. This should be changed to “The in vitro clots used in this study were homogenous, which may not be representative…” because the “and” after the comma is unnecessary.

Reviewer #2: This manuscript describes the effects of rt-PA dose on histotripsy-combined, catheter-directed thrombolytic efficacy in vitro.

1. The designed test schemes and procedures seem technically sound and clearly described. Sufficient data support the conclusion that histotripsy can reduce the dose of rt-PA infusion via catheter by a factor of twenty compared to the standard dose for the equivalent thrombolytic efficacy.

2. Statistical analyses conducted in this study seem appropriate. Data interpretation seems unbiased and reasonable without misleading explanation.

3. All data underlying the findings in the manuscript are available as an excel file and jpg figure files on the journal website.

4. The manuscript contains organized contents written in standard English.

5. (Additional comments) Some minor revisions seem required before publication. The itemized comments are below:

Page 3-4, introduction:

The research motivation can be more specific for catheterized conditions. Many previous works already demonstrated that ‘rt-PA + histotripsy’ improves sonothrombolytic efficacy and further reduces the rt-PA dose for the equivalent clot mass reduction. Although the novelty of this work is catheter-directed rt-PA delivery instead of systematic venous administration, the given hypothesis ‘histotripsy may reduce the required rt-PA dose’ seems not significant to justify the further experimental study since the overall trend of the results can easily be anticipated based on the previously published works. The authors may emphasize 1) what is the specific benefit of combining histotripsy to catheter-directed lytic infusion in comparison with standard venous administration and 2) what kinds of obstacles or advantages can uniquely involve in the catheter-directed environment, e.g. if catheter tip may provide an impedance boundary that is different from free field cavitation, if ~200 ul/min infusion-induced local streaming and associated local agitation may help or hinder nucleation, etc. Some of these points are briefly discussed in the Discussion, but it would be more helpful to follow the research motivation by clarifying them in the Introduction.

Page 5, lines 92-93:

The frequency of the used histotripsy pulses should be provided.

Page 7, lines 136-137:

Please add the information of duty-cycle and corresponding time-averaged acoustic intensity.

Does ‘13.3 us for 20 cycle’ burst mean that the frequency is 1.5 MHz (1/(13.3e-6/20))? Please clarify the frequency so the readers easily catch the specifications and estimate the 1) frequency-dependent focal volume, 2) scattering, attenuation through the propagation, and 3) cavitation cloud dynamics.

Reviewer #3: The authors describe use of histotripsy and rt-PA for treatment of human blood clots, with the goal of utilizing histotripsy while reducing rt-PA dose to increase safety. This work addresses a significant problem with potential for significant impact. The work is well performed and described, limitations are clearly explained. The discussion is well considered, especially catheter effects and methods of clot dissolution, and was interesting to read. I have only a few minor comments.

Specific comments:

1. “Clots were stored at 4 ℃ for 3 days to allow for full retraction [22], and were used within 2 weeks of formation.”

Alternatively, if aged clots were used, what would be the expected effect on experiments?

2. “A multi-side port, 5F catheter (Cragg-McNamara, Medtronic, Minneapolis, Minnesota, USA) was introduced into the model vessel 104 via a hemostasis valve and positioned approximately through the central axis of the clot. No guidewires or sheaths were used to advance the catheter through the clot.”

Would there be any risk of downstream embolization due to mechanical contact with the clot?

3. What factors must be considered before using one of these strategies for in vivo treatment? For example, how will the transducer be coupled to the patient and focused at the correct location? Is 21 min an acceptable treatment time considering both the urgency of recanalization and potential for motion/registration errors during this time? Are there concerns regarding heating at the skin?

4. Some internal reference/bookmark errors need to be corrected: “The generation of D-dimer is reported in Figure Error! No text of specified style in document”. The figure before this is Fig. 4 and the figure after this is Fig. 5, so it’s unclear if figures are numbered correctly.

5. “Regardless, these data indicate that histotripsy bubble clouds can be initiated in conjunction with catheter-directed thrombolysis.”

In this location, can the authors remind the reader of the significance of why it is important that these can be done in conjunction?

6. “The data in this study indicate fibrinolysis is equivalent between the standard catheter-directed thrombolysis dosing scheme and a twenty-fold reduction in rt-PA when combined with histotripsy (Figure 5). A reduction in lytic dose may improve the treatment safety profile [9,21].”

This is true that reducing t-PA dose could improve one component of safety, but what are the safety considerations of histotripsy at peak acoustic pressures of 35 MPa?

6. PLOS authors have the option to publish the peer review history of their article (what does this mean?). If published, this will include your full peer review and any attached files.

Reviewer #1: No

Reviewer #2: **Yes: **Jinwook Kim

Reviewer #3: No

---

## [Author Response · Author response to Decision Letter 0]

24 Nov 2021

We appreciate the constructive comments provided by reviewers, and have modified the text accordingly. Below, reviewers’ comments are in listed, along with our responses, and modified portions of the text. 

Reviewer #1: This paper reports an interesting sonothrombolysis by using catheter infusion and histotripsy. The findings on reduced tPA is promising. The following comments may be addressed to strengthen the manuscript:

1. Abstract

Page 2. Line 29. “… rt-PA at a one-twentieth dose reduction (0.23 μg/mL) when combined with histotripsy.”. This should be changed to “… rt-PA at a reduction to one-twentieth of the common clinic dose (0.23 μg/mL) when combined with histotripsy.”.

We have updated the text:

Line 30: “…rt-PA at a reduction to one-twentieth of the common clinical dose (0.23 μg/mL) when combined with histotripsy.”

Page 2. Line 36. “Overall, the data collected in this study indicate the rt-PA dose can be reduce by…”. This should be changed to “Overall, the data collected in this study indicate the rt-PA dose can be reduced by…”. (use past tense for “reduce”)

We have updated the text as requested:

Line 39: “Overall, the data collected in this study indicate the rt-PA dose can be reduced by…”

2. Introduction

Page 3. Line 49. ” This approach is not without…”. This could be changed to “This approach has…” to be more concise.

We have revised the text:

Line 52: “This approach has…”

Page 3 Line 55. Authors may add a paragraph here to introduce the status of sonothrombolysis including the ultrasound catheter based sonothrombolysis (e.g. EKOS, forward-viewing catheter based sonothrombolysis), focused ultrasound based sonothrombolysis, etc. , and their limitations, and then come to the uniqueness of the histotripsy based sonothrombolysis that is reported in this paper.

The following text was added:

Line 58: “Sonothrombolysis relies on the ultrasound energy to improve the activity of rt-PA [12]. Ultrasound can be administered intravascularly (e.g. EKOS Endovascular System, Boston Scientific, Marlborough, MA [13] and forward facing catheters [14]) or extracorporeally [15,16]. Bubble activity is the primary catalyst to promote clot degradation [12]. Most sonothrombolysis schemes employ exogenous agents such as microbubbles, nanobubbles, or droplets to nucleate bubble activity [17,18]. These methods have a limited thrombus penetration profile [19] and do not sustain bubble activity for extended periods (< 30 s) [20,21]. Histotripsy is a focused ultrasound therapy that achieves tissue ablation via the mechanical activity of bubble clouds [22–24]. The tension of the histotripsy pulse is sufficient to activate nanoscale nuclei that permeate soft tissues such as clot [25,26], enabling bubble generation throughout the thrombus burden.”

3. Methods and materials

Page 5 Line 74-81. Authors may comment on the clot retraction. Are they retracted clots or unretracted ones?

The following text was added to address clot retraction:

Line 97: “At the conclusion of incubation period, clots were retracted from the container walls and significant serum was observed in the pipette.”

Page 6. Line 109. “Bubble activity was initiate in the clot…”. This should be changed to “Bubble activity was initiated in the clot…” so that “was” and “initiated” are both past tense.

We have revised the text as request:

Line 131: “Bubble activity was initiated in the clot…”

Page 7. Line 129. “Motor positions that contained bubble activity within the clot were assigned in 5 mm increments along the length…”. I’m not sure if it is correct to say that a motor position can contain bubble activity. How about this: “Motor positions that correspond with bubble activity within the clot were assigned in 5 mm increments along the length…”.

The text was revised based on the reviewer’s recommendation:

Line 151: “Motor positions that aligned the histotripsy focus within the clot were determined in 5 mm increments along the length of the clot (3 in total).….”

Page 7. Line 139. “…scanned the histotripsy source scanned along the length…”. Remove the second “scanned” so that this phrase is written as “…scanned the histotripsy source along the length...”.

The text was revised as requested:

Line 171: “The multi-path scheme also scanned the histotripsy source…”

Page 8. Line 156. “A total of eight clots were tested for each arm, and a total of 72 total clots used in the study.”. This should be changed to: “Eight clots were tested for each arm, with a total of 72 clots used in the study.” to avoid repeating “total” 3 times.

The text was revised as requested:

Line 192: “Eight clots were tested for each arm, with a total of 72 clots used in the study.”

Page 8. Line 158. “…the residual clot was removed from the flow model and was weighed…”. The second “was” can be removed to yield: “…the residual clot was removed from the flow model and weighed…”.

The text was updated accordingly:

Line 194: “…the residual clot was removed from the flow model and weighed…”

Page 8. Line 161. “The instrument was examination with brightfield microscopy…”. For this sentence, “examined” should be used. The sentence becomes: “The instrument was examined with brightfield microscopy…”.

We revised the text as requested:

Line 197: “The instrument was examined…”

4. Results

Page 11. Line 232. “Hemoglobin, intact erythrocytes, and intact platelets shed from the clot are reported for all treatment arms in”. This sentence/section needs to be completed.

Page 12. Line 241. There was already a “Figure 4” mentioned previously.

For both these comments, there appears to have been an issue with the rendering of our text. The paragraph reads as follows:

Line 276: “Hemoglobin, intact erythrocytes, and platelets in the plasma perfusate following treatment are reported for all arms in Figure 4. In the absence of rt-PA, histotripsy increased hemoglobin production relative to control (no histotripsy). For arms with rt-PA, there was no difference in the hemoglobin concentration in the plasma for arms with or without histotripsy exposure (single- or multi-path histotripsy insonation schemes). For a given lytic concentration, no differences were observed in the concentration of red blood cells for arms with or without histotripsy (single- and multi-path insonation scheme). Similarly, neither histotripsy insonation scheme (single or multi-path) had an influence on platelets shed from the clot.” 

Page 12. Line 248. “Figure Error! No text of specified style in document.”. Add the figure number.

We have added the appropriate figure number:

Line 299: “…in Figure 5…”

Page 12. Line 253. “…histotripsy significantly increased D imer…”. Change “D imer” to “D-dimer” so that the phrase becomes “…histotripsy significantly increased D-dimer…”.

The text was revised:

Line 303: “…significantly increased D-dimer…”

Page 13. Line 257. “Figure Error! No text of specified style in document.:”. Add the figure number.

We have added the figure number

Line 310: “Figure 5…”

Page 13. Line 268. “The multi-path scheme was similarly skewed FDP production to higher concentrations compared to lytic alone.”. Change this to: “Similarly, the multi-path scheme skewed FDP production to higher concentrations compared to lytic alone.”.

The text was revised:

Line 320: “Similarly, the multi-path scheme skewed…”

Page 15. Line 299. “The average Euclidean distance between acoustic emissions maxima and catheter location…”. Change “emissions” to “emission’s” so that it is possessive. The phrase becomes: “The average Euclidean distance between the acoustic emission’s maxima and catheter location…”.

The use of “emission’s” is too colloquial for a technical journal such as PLoS ONE. Further, given that there are multiple emissions, this should technically be changed to “emissions’”. We revised the text in the spirit of the comment:

Line 357: “The average Euclidean distance between the maxima of acoustic emission and catheter location…”

5. Discussion

Page 18. Line 357. “The histotripsy pulse will be attenuated as it propagated through the clots…”. Change “propagated” to “propagates” so that the tenses in the sentence correspond chronologically. The phrase becomes: “The histotripsy pulse will be attenuated as it propagates through the clots…”.

The text has been updated:

Line 428: “The histotripsy pulse will be attenuated as it propagates through the clot…”

Page 20. Line 415. “…the release of non-specific fibrin degradation products was increased for histotripsy arms at two highest lytic dose…”. This should be changed to “…the release of non-specific fibrin degradation products was increased for histotripsy arms at the two highest lytic doses…” such that “dose” is switched to plural.

The text was updated:

Line 497: “…the release of non-specific fibrin degradation products was increased for histotripsy arms at the two highest lytic doses.”

Page 21. Line 436. “The in vitro clots used in this study were homogenous, and which may not be representative…”. This should be changed to “The in vitro clots used in this study were homogenous, which may not be representative…” because the “and” after the comma is unnecessary.

We have revised the text as requested:

Line 521: “The in vitro clots used in this study were homogenous, which….”

Reviewer #2: This manuscript describes the effects of rt-PA dose on histotripsy-combined, catheter-directed thrombolytic efficacy in vitro.

1. The designed test schemes and procedures seem technically sound and clearly described. Sufficient data support the conclusion that histotripsy can reduce the dose of rt-PA infusion via catheter by a factor of twenty compared to the standard dose for the equivalent thrombolytic efficacy.

2. Statistical analyses conducted in this study seem appropriate. Data interpretation seems unbiased and reasonable without misleading explanation.

3. All data underlying the findings in the manuscript are available as an excel file and jpg figure files on the journal website.

4. The manuscript contains organized contents written in standard English.

5. (Additional comments) Some minor revisions seem required before publication. The itemized comments are below:

Page 3-4, introduction:

The research motivation can be more specific for catheterized conditions. Many previous works already demonstrated that ‘rt-PA + histotripsy’ improves sonothrombolytic efficacy and further reduces the rt-PA dose for the equivalent clot mass reduction. Although the novelty of this work is catheter-directed rt-PA delivery instead of systematic venous administration, the given hypothesis ‘histotripsy may reduce the required rt-PA dose’ seems not significant to justify the further experimental study since the overall trend of the results can easily be anticipated based on the previously published works. The authors may emphasize 1) what is the specific benefit of combining histotripsy to catheter-directed lytic infusion in comparison with standard venous administration and 2) what kinds of obstacles or advantages can uniquely involve in the catheter-directed environment, e.g. if catheter tip may provide an impedance boundary that is different from free field cavitation, if ~200 ul/min infusion-induced local streaming and associated local agitation may help or hinder nucleation, etc. Some of these points are briefly discussed in the Discussion, but it would be more helpful to follow the research motivation by clarifying them in the Introduction.

We have added the following text to address this point:

Line 77: “Prior histotripsy-enhanced thrombolysis studies modeled a systemic rt-PA infusion, whereas catheter-directed therapy is the standard-of-care for DVT [6]. There are multiple differences between systemic and catheter-directed therapies that may alter treatment outcomes, including the lytic profile and administration rate. Further, the influence of the catheter on histotripsy bubble nucleation and dynamics is unknown.”

Page 5, lines 92-93:

The frequency of the used histotripsy pulses should be provided.

We have included additional specifications of the transducer:

Line 110: “Histotripsy pulses were generated by an eight-element, spherically-focused transducer with an elliptical geometry (9 cm major axis, 7 cm minor axis) and 1.5 MHz fundamental frequency.”

Page 7, lines 136-137:

Please add the information of duty-cycle and corresponding time-averaged acoustic intensity.

Does ‘13.3 us for 20 cycle’ burst mean that the frequency is 1.5 MHz (1/(13.3e-6/20))? Please clarify the frequency so the readers easily catch the specifications and estimate the 1) frequency-dependent focal volume, 2) scattering, attenuation through the propagation, and 3) cavitation cloud dynamics.

This is a good issue to raise. Calibration of the source at the driving levels employed in this study is not possible due to bubble formation confounding the measurement. Hence we do not have access to the full pressure waveform required to compute the time-averaged intensity. To clarify this point, the following text was added:

Line 163: “At each treatment location, 2,000 pulses of 13.3 µs pulse duration (nominally 20 cycles of the 1.5 MHz fundamental driving frequency) were applied at a 40 Hz rate (0.05 % duty cycle). The focal pressure was calibrated for driving levels up to ~25 MPa. Bubble formation prevented calibration for larger driving levels, so the peak negative pressure level was estimated by linear extrapolation based on the available calibration as described previously [41]. Time-averaged intensities for histotripsy are typically < 100 W/cm2 because of the low duty cycle [25].” 

Reviewer #3: The authors describe use of histotripsy and rt-PA for treatment of human blood clots, with the goal of utilizing histotripsy while reducing rt-PA dose to increase safety. This work addresses a significant problem with potential for significant impact. The work is well performed and described, limitations are clearly explained. The discussion is well considered, especially catheter effects and methods of clot dissolution, and was interesting to read. I have only a few minor comments.

Specific comments:

1. “Clots were stored at 4 ℃ for 3 days to allow for full retraction [22], and were used within 2 weeks of formation.” Alternatively, if aged clots were used, what would be the expected effect on experiments?

This is a good question. Clots formed in vitro are typically homogenous compared to those form in vivo, and do not contain some of the elements observed for aged venous thrombi. The following text was added to address this point:

Line 100: “Clots formed under this protocol are uniformly homogenous with erythrocytes [36], which may not reflect the heterogeneous pathologies observed for aged in vivo specimen [10].”

2. “A multi-side port, 5F catheter (Cragg-McNamara, Medtronic, Minneapolis, Minnesota, USA) was introduced into the model vessel 104 via a hemostasis valve and positioned approximately through the central axis of the clot. No guidewires or sheaths were used to advance the catheter through the clot.” Would there be any risk of downstream embolization due to mechanical contact with the clot?

Possibly, though catheter infusion is part of the standard of care, and deemed an acceptable risk. Interventionalists may place a filter distal to the catheter (e.g. between the catheter and heart) to collect embolized debris. The following text was added: 

Line 124: “To be consistent with current treatment strategies for DVT [6], rt-PA was administered via a multi-side port, 5F catheter (Cragg-McNamara, Medtronic, Minneapolis, Minnesota, USA). The catheter was introduced into the model vessel via a hemostasis valve and positioned approximately through the central axis of the clot.”

Line 517: “Because of embolization risk during CDT, interventionalists may place a filter in the inferior vena cava [64]. The filter may alter the flow profile of venous return, whereas here a fixed flow rate was used.”

3. What factors must be considered before using one of these strategies for in vivo treatment? For example, how will the transducer be coupled to the patient and focused at the correct location? 

This is a good question. We added the following text to address this point: 

Line 523: “Here, studies were conducted in a tank of degassed water to couple the acoustic field to the clot. A bath of degassed water with an acoustic window will be required to target the clot in vivo, and the ultrasound pulse will have to transverse several centimeters of tissue [15, 65]. Three orthogonal motorized stages were used to translate the histotripsy source, which is effective for the controlled setup in this in vitro study. Additional flexibility for transducer placement will be required for treatment in vivo. One possible solution to distribute bubble activity throughout the clot burden are collaborative robots, which have six degrees-of-freedom to enable arbitrary angulation of the histotripsy source relative to the target [53].”

Is 21 min an acceptable treatment time considering both the urgency of recanalization and potential for motion/registration errors during this time? 

Good question. We added the following text to address this point:

Line 467: “Catheter-directed thrombolytics are administered over the course of several days [9], much longer than the treatment time considered here (~ 21 min). We anticipate it is possible to optimize the histotripsy insonation scheme and reduce the treatment duration, minimizing the potential for off-target effects. Even for the unoptimized scheme tested here, the results in this study indicate histotripsy could significantly reduce the overall duration required for rt-PA administration.”

Are there concerns regarding heating at the skin?

Because of the low duty cycle (0.05% here), there are no observed effects in terms of tissue heating. The following text was added to address this issue:

Line 69: “The duty cycle for histotripsy pulsing schemes is very low (< 1%), making tissue heating effects negligible [27].”

4. Some internal reference/bookmark errors need to be corrected: “The generation of D-dimer is reported in Figure Error! No text of specified style in document”. The figure before this is Fig. 4 and the figure after this is Fig. 5, so it’s unclear if figures are numbered correctly.

We have reviewed the figure references, and confirmed they are listed correctly. 

5. “Regardless, these data indicate that histotripsy bubble clouds can be initiated in conjunction with catheter-directed thrombolysis.” In this location, can the authors remind the reader of the significance of why it is important that these can be done in conjunction?

The following text was added:

Line 393: “Histotripsy bubble activity must act concurrently with rt-PA administration to improve fibrinolysis [33]. The data collected here indicate that histotripsy bubble clouds can be initiated in conjunction with CDT to improve rt-PA activity, and therefore treatment outcomes.”

6. “The data in this study indicate fibrinolysis is equivalent between the standard catheter-directed thrombolysis dosing scheme and a twenty-fold reduction in rt-PA when combined with histotripsy (Figure 5). A reduction in lytic dose may improve the treatment safety profile [9,21].” This is true that reducing t-PA dose could improve one component of safety, but what are the safety considerations of histotripsy at peak acoustic pressures of 35 MPa?

The following text was added:

Line 401: “A reduction in lytic dose may reduce the likelihood of bleeding associated with rt-PA administration [9,34]. Risks associated with histotripsy may include off-target damage, though these effects can be minimized using appropriate sources and pulsing schemes [39]. Histotripsy exposure may also result in regional endothelial denudation [50], consistent with the damage profile of current clinical thrombectomy devices [51].”

---

## [Decision Letter · Decision Letter 1]

6 Dec 2021

(More Than) Doubling Down: Effective fibrinolysis at a reduced rt-PA dose for catheter-directed thrombolysis combined with histotripsy

PONE-D-21-29418R1

Dear Dr. Bader,

We’re pleased to inform you that your manuscript has been judged scientifically suitable for publication and will be formally accepted for publication once it meets all outstanding technical requirements.

Kind regards,

Virginie Papadopoulou, Ph.D.

Academic Editor

PLOS ONE

Additional Editor Comments (optional):

Dear authors,

Thank you for your careful revision. All comments raised have been addressed. It is my pleasure to accept the manuscript for publication.

Sincerely,

Virginie Papadopoulou, PhD

Reviewers' comments:

Reviewer's Responses to Questions

**Comments to the Author**

1. If the authors have adequately addressed your comments raised in a previous round of review and you feel that this manuscript is now acceptable for publication, you may indicate that here to bypass the “Comments to the Author” section, enter your conflict of interest statement in the “Confidential to Editor” section, and submit your "Accept" recommendation.

Reviewer #1: All comments have been addressed

Reviewer #2: All comments have been addressed

Reviewer #3: All comments have been addressed

2. Is the manuscript technically sound, and do the data support the conclusions?

Reviewer #1: Yes

Reviewer #2: Yes

Reviewer #3: Yes

3. Has the statistical analysis been performed appropriately and rigorously? 

Reviewer #1: Yes

Reviewer #2: Yes

Reviewer #3: Yes

4. Have the authors made all data underlying the findings in their manuscript fully available?

Reviewer #1: Yes

Reviewer #2: Yes

Reviewer #3: Yes

5. Is the manuscript presented in an intelligible fashion and written in standard English?

Reviewer #1: Yes

Reviewer #2: Yes

Reviewer #3: Yes

6. Review Comments to the Author

Reviewer #1: (No Response)

Reviewer #2: The responses with a revised manuscript have addressed the previous comments. I do not have any other questions or comments.

Reviewer #3: The authors have addressed the previous comments adequately.

Suggest changing "...transverse several centimeters of tissue..." to "..traverse several centimeters of tissue..." in the revised manuscript.

7. PLOS authors have the option to publish the peer review history of their article (what does this mean?). If published, this will include your full peer review and any attached files.

Reviewer #1: No

Reviewer #2: No

Reviewer #3: No

---

## [Editor Report · Acceptance letter]

23 Dec 2021

PONE-D-21-29418R1 

(More Than) Doubling Down: Effective fibrinolysis at a reduced rt-PA dose for catheter-directed thrombolysis combined with histotripsy 

Dear Dr. Bader:

I'm pleased to inform you that your manuscript has been deemed suitable for publication in PLOS ONE. Congratulations! Your manuscript is now with our production department. 

Kind regards, 

on behalf of

Dr. Virginie Papadopoulou 

Academic Editor

PLOS ONE